# Prescribed Burning as A Management Tool to Destroy Dry Seeds of Invasive Conifers in Heathland in Denmark

**Christian Andreasen** [1,*], **Kasper Rossing** [1] **and Christian Ritz** [2]

1   Department of Plant and Environmental Sciences, University of Copenhagen, Højbakkegaard Allé 13, DK-2630 Taastrup, Denmark; info@landcare.dk
2   Department of Nutrition, Exercise and Sports, Faculty of Science, University of Copenhagen, Rolighedsvej 25, DK-1958 Frederiksberg C, Denmark; ritz@nexs.ku.dk
*   Correspondence: can@plen.ku.dk; Tel.: +45-5132-2551

**Abstract:** Non-indigenous conifers are considered invasive to the coastal dune heathland in Denmark, and massive clearing is carried out in an attempt to recreate and keep the original heathland. Burning is a common method for managing, but its feasibility to control the seed bank of conifers has not been investigated. This project shows that the burning of logged conifer trees will often eliminate seeds of lodgepole pine, mugo pine and Sitka spruce, even when the seeds were placed into a depth of five centimeters in the soil. The effect on seeds depends on the fuel load and the fire conditions (e.g., dryness, wind, and temperature). If the seeds were exposed to a high temperature, the seeds were not able to germinate afterward. The temperature was about 80 °C for all species. If the sum of temperatures based on temperature records every 30 s exceeded between 12,000 and 14,000 °C no seeds were able to germinate. The relationship between the mean temperature of the burns and the germination rate at seeds placed in various soil depths was modelled. Findings should be interpreted cautiously as each depth-species combinations were not replicated in space or time due to practical constraints.

**Keywords:** lodgepole pine; land restoration; landscape fire; mugo pine; *Pinus mugo*; *Pinus contorta*; Picea sitchensis; Sitka spruce; thermal weed control

## 1. Introduction

Burning is currently used as a management tool to manage ecosystems [1]. Control of invasive plant species by burning has been used in several countries for decades, utilizing the native floristic adaptation to regular bushfires and lack of fire resistance of the invasive weeds [2,3]. Prescribed burning is often considered a sustainable management tool in land restoration because fires occur naturally in dry landscapes and contribute to the sustainable development of natural ecosystems; meaning a development that meets the needs of the present without compromising the ability of future generations of native species to meet their needs.

However, sometimes, burning favours the invasive weed species [4,5]. Numerous studies have shown the influence of burning on the regeneration ability of shrubs, herbaceous species and trees [5–8]. High temperatures may destroy seeds or may physically break the dormancy of plant species with a hard seed coat [8,9]. Unharmed seeds buried in the soil may be affected by heat and smoke, either directly or by the substances dissolved by rains after a fire, which stimulate or inhibit the germination process [10–12]. Invasive conifers have become an increasing problem on Danish heathlands, and prescribed burning is used to overcome the problem, but the effect of burning on the soil seed bank is unknown. In other parts of the world, invasive pines have also become a threat to natural ecosystems.

In South Africa, an attempt to controlled invasive pines with prescribed burning has shown limited success [13,14].

The coastal dune heathland of northern Denmark represents a unique type of nature dominated by perennial subshrubs (e.g., *Calluna vulgaris* L. Hull, *Erica tetralix* L. and *Salix repens* L.) and native grasses and sedges. Non-indigenous conifers were introduced to the Danish heathlands in the late eighteenth century to establish plantations and control the sand drift which had caused severe problems for the farming community, the infrastructure, and the inhabitants in general [15]. The introduction of the mugo pine (*Pinus mugo* Turra) was a breakthrough in stopping the sand drift. Other less hardy but more productive conifer species were also introduced like the American lodgepole pine (*Pinus contorta* Douglas ex Loudon) and Sitka spruce (*Picea sitchensis* (Bong.) Carriére) resulting in the establishment of plantations with a size up to 1400 ha [15].

The integration of the Sand Drift Law with the Nature Protection Act in 1992 [16] in combination with a significant decline in the occurrence of many bird species belonging to the heathland was an eye-opener in recognizing the invasive properties of the introduced conifers. Additionally, the fact that the plantations generated little or no profit was an incentive towards a change of attitude, which led to reestablishing the original heathland in many places [17,18].

Different management tools are used to control the conifer of which logging, followed by chipping or crushing, is the most common. Chipping or logging, however, requires the use of heavy machinery leaving the possibility of damaging the dunes and compress the soils on wet areas. However, conifer trees often reestablish after the areas have been cleared, probably caused by germinated seeds from the soil seed bank.

Burning is a well-known tool in heath management in Denmark [17,19], but no scientific information is available about burning as a management tool to control seeds of conifers on heathlands in Denmark.

Only scarce information is available about the viability and seed longevity of the three conifer species. The viability of seeds of the lodgepole pine in soils is not known, but stored at a temperature of 0−5 °C the seeds can be viable for more than 7 years [20] and up to 17 years in cold storage [21]. The longevity of seeds of mugo pine and Sitka spruce in soils have yet to be investigated but in ordinary storage, unsealed and with no temperature control, mugo pine seeds can be viable 1–2 years [20].

Our research aimed to investigate how the germination ability of the seeds of lodgepole pine, Sitka spruce and mugo pine, placed on the ground and in the uppermost part of the soil layer, was affected by burning. We investigated whether seeds were killed, harmed, maintained their germination ability or improved germination ability after being exposed to the standard practice of burning logged conifer trees on heathland.

## 2. Materials and Methods

### 2.1. Study Area

The experimental area was a coastal dune heath located on Råbjerg Mile approximately 17 km southwest of Skagen the northernmost town in Denmark. The area was mainly covered with *P. mugo* trees, which were logged and left on site the previous year to spot fire the trees. Wooden poles and string fenced off the area. A wood chipper vehicle cleared the surrounding area of pines creating firebreaks and formed a good overview and access for setting up experiments. Eight independent experiments were completed in the period from 18 October to 11 April with permission from the Forest and Nature Agency, Danish Ministry of the Environment. The eight experimental sites were selected on the criteria of having as even a distribution of pine as possible.

On each experimental day, a site was selected depending on wind conditions. In a moderate breeze, a low-lying site was selected on the leeward side of a sand dune. In calm weather, the site could be located on a higher elevation (e.g., on the side and/or on the top of a sand dune). Bio-diesel was sprayed with a spray bottle on the fuel to promote ignition. The fire was lit at the side of the fuel

pile opposite to the wind direction creating a headwind fire. All experiments were carried out under different conditions (fuel load, soil and weather conditions) resulting in a great variation of humidity, smoke, wind, temperature range and duration of burnings (Figure 1B) reflecting standard practice.

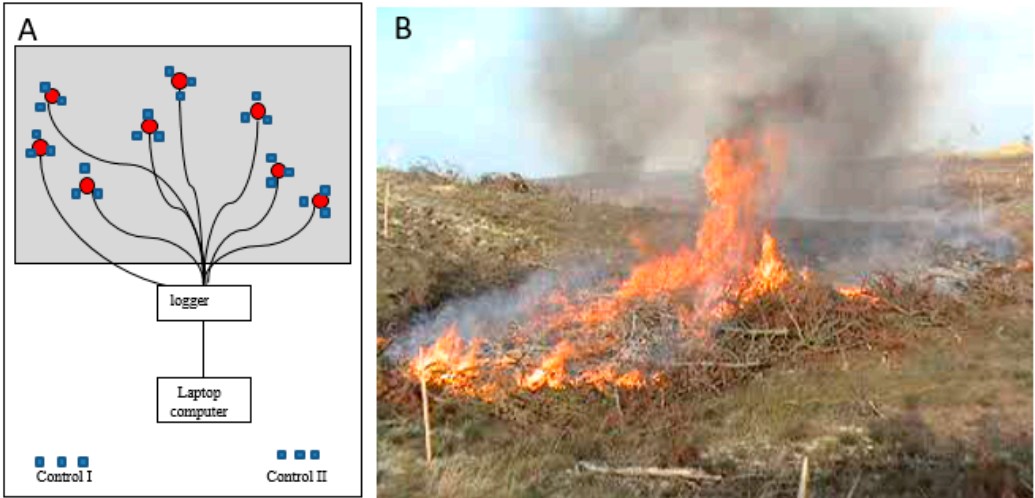

**Figure 1.** (**A**). Illustration of the set-up of an experiment seen from above. The grey area is where the fire burned. Eight thermocouple joints (temperature sensors) (●) were placed on a site surrounded by three nets with seeds (■) all in the same depths. In each of the three nets, there were one plant species. All 24 nets were placed at the same level (e.g., on the soil surface or buried in the ground). (**B**). A burning site.

## 2.2. Seeds and Seed Nets

Seed used for the experiments were randomly taken from a seed lot of each conifer species delivered from the Danish Forest and Nature Agency. Seed nets were used to ensure that the seed could be found after the burning. The nets were fire-resistant up to 900 °C, allowing the seeds to have contact with the surroundings and avoiding that the nets could protect the seeds from thermal shock (Figure 2). The nets were made of steel fly net electro-galvanized with zinc, measuring 5 cm × 5 cm, folded and stapled along the edges for reinforcement. A blast-heater heated up the nets until they became red-hot, eliminating the risk of seeds being affected by harmful emission from the material. Subsequently, the nets were rapidly cooled in water to release the zinc. Seeds were placed in one layer avoiding that seeds protected each other from the heat.

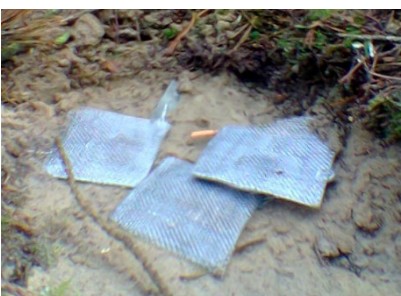

**Figure 2.** Seed nets containing one layer of seeds.

## 2.3. Experimental Design

Eight experiments were conducted. Each experiment was placed on an area of approx. 5 m × 5 m (Figure 1B). In each experiment, 24 nets with seeds were placed in the same depth (8 nets for each plant species) corresponding to the eight spots in Figure 1A. Only one depth was monitored at each

burning site. At three sites, seeds were placed at the soil surface. At the five other sites, the seeds were placed in 1, 2, 3, 4, and 5 cm depth, respectively (Table 1). Each net contained 100 seeds of one plant species. After placing the seeds in the nets, the nets were sealed and tagged by a piece of a steel band. Information about the plot, sensor number and name of the plant species were punched into the tag with a plunger.

**Table 1.** Description of the sites, date of fire and location of the seeds in the soil.

| Type of Heather | Site | Date of Fire | Location of the Seeds (Depth in cm) |
|---|---|---|---|
| Sand dune heather with logged conifers | 1 | 28 October | 0 |
| Sand dune heather with logged conifers | 2 | 18 November | 1 |
| Sand dune heather with logged conifers | 3 | 13 December | 2 |
| Sand dune heather with logged conifers | 4 | 16 December | 3 |
| Sand dune heather with logged conifers | 5 | 5 April | 4 |
| Sand dune heather with logged conifers | 6 | 6 April | 5 |
| *Calluna* heather | 7 | 10 April | 0 |
| *Calluna* heather | 8 | 11 April | 0 |

Thermocouples joints (temperature sensors) were placed in the middle of three seed nets and in the same depth (Figure 1A). They were constructed of two wires conjoined by a copper tube tested and approved by the Danish Institute of Fire and Security to measure temperatures up to 1000 °C. The thermocouple wires were placed in canals covered with sand to protect them from the heat. Each spot was marked with a painted iron rod. The thermocouples were connected to a USB TC-08 Thermocouple Data Logger (Pico Technology Limited, Cambridgeshire, United Kingdom) which was connected to a computer. The data logger was equipped with eight channels to measure temperatures in the eight places during the experimental burning (Figure 1A) to determine the temperature distribution, which varied significantly between spots due to an uneven distribution of fuel (Figure 1B). All seed nets and sensors were placed at the same depth in an experiment. Two seed nets with 100 seeds of each species were placed 15 m outside the burning and smoky area as controls in the same depth as the seeds inside the burning area close to the data logger (Figure 1A). The temperatures at the control seeds were measured with thermometers during the burning.

Table 1 shows how deeply the seeds were buried in the experiments. Six fires were set on sites with logged conifers, and two fires were set on sites with *Calluna vulgaris* heather. Experiments 7 and 8 received the same treatment, but the heather at experiment 8 had less biomass resulting in lower temperatures of the fire.

The duration of the burning was recorded, and the sum of temperatures based on temperature records every 30 s was calculated. All experiments were different because the fuel load and the fire conditions varied (e.g., dryness and wind), ensuring a wide range of temperatures which was essential for the modelling.

After the wood was burned, seed nets were removed from the spot and stored in plastic bags in a refrigerator at 8 °C until the germination ability of the seeds was tested.

*2.4. Germination Test*

One hundred seeds of each conifer species from each of the eight burning experiments plus two hundred control seeds (Figure 1A) were tested for germination. A hundred seeds were placed on filter paper (type AGF 138, particle retention (liquid) 15 µm, weight 90 g m$^{-2}$, produced by Frisenette Aps, Knebel, Denmark) saturated with water. The filter paper was folded as an envelope. The seed envelopes were placed on a filter saturated with water. The paper was folded, rolled up, put into a plastic bag, and placed in an incubator at 23 °C and exposed to light.

Seeds of Sitka spruce were pre-treated by storing the moisturized seed rolls in a refrigerator for three weeks at 8 °C, which is usual practice for this species to promote germination [22,23].

Germination was monitored continuously by opening the seed rolls. A seed was assumed germinated when the radicle had obtained the same length as the seed. Germinated seeds were counted and removed at every reading for four weeks.

*2.5. Statistical Analysis*

Germination data were analyzed using logistic mixed-effects regression models as the response of each seed was binary (germinated/not germinated). Germination probabilities were modelled in terms of log odds. Specifically, the probability may be calculated from the log odds using the formula exp(log odds)/(1 + exp(log odds)). Fixed and random effects were specified on the log odds scale. A single model was fitted to all data. Specifically, we assumed the following quadratic relationship between the log odds and the logarithm-transformed mean temperature:

$$\alpha_1(depth, species) + \alpha_2(species) \cdot logtemperature + \alpha_3(species) \cdot (logtemperature)^2 + a_{spot} \qquad (1)$$

with $\alpha_3 < 0$ ensuring that germination probabilities approach 0 for temperatures approaching 0 °C or getting very large. As indicated in Equation (1), the intercept parameter $\alpha_1$ was assumed to differ between 3 depths (grouped as 0–4 cm, 5 cm, and control) and 4 species, whereas the linear and quadratic coefficients $\alpha_2$ and $\alpha_3$ were assumed to differ between species only. Observations from 0–4 cm were put in the same group due to the limited number of germinated seeds in these soil layers. The experiments were carried out under different conditions (fuel load, soil and weather conditions), resulting in considerable variation of humidity, smoke, wind, temperature range and duration of burnings between spots. This variation was modeled by including spot-specific random intercept $a_{spot}$, which were assumed to follow mean-zero normal distribution with unknown standard deviation $\sigma_{spot}$. Estimation was based on the maximum likelihood principle. The statistical environment R was used for the analysis [24] and in particular, the R add-on packages "car" [25] and "lme4" [26].

## 3. Results

All experiments were different because the fuel load and the fire conditions varied resulting in a wide range of temperatures, which was essential for the modelling. Figure 3 shows an example of a course of temperatures during a burning (28 October). Seeds placed outside the fire (controls) were only exposed for a small variation in temperatures. Maximum, mean, and sum of temperatures at each of the eight seed locations are summarized in Table 2 and Figure 4.

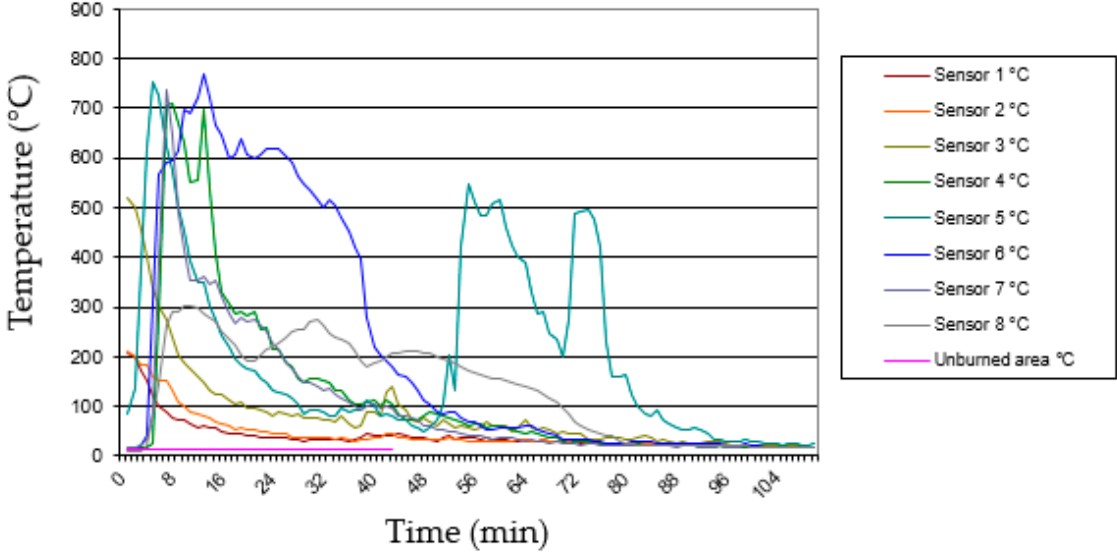

**Figure 3.** Course of temperatures at the soil surface where the sensors and nets with weed seeds were placed during the fire, 28 October. Maximum temperatures between 205 °C to 768 °C.

**Table 2.** Maximum, mean and sum of temperatures at each of the eight seed locations in the eight fires and the germinations percentages tested afterward.

| Date (Seed Location) | Plot No. | Max. Temp. (°C) | Mean Temp. (°C) | Sum of Temp. (°C) | Germination Percentages | | |
|---|---|---|---|---|---|---|---|
| | | | | | *P. mugo* | *P. contorta* | *P. sitchensis* |
| 28 October | 1 | 205 | 40.2 | 8834 | 0 | 0 | 0 |
| (0 cm) | 2 | 212 | 44.9 | 9874 | 0 | 0 | 0 |
| | 3 | 521 | 86.3 | 18,974 | 0 | 0 | 0 |
| | 4 | 708 | 122.7 | 26,978 | 0 | 0 | 0 |
| | 5 | 751 | 202.9 | 44,632 | 0 | 0 | 0 |
| | 6 | 768 | 218 | 47,966 | 0 | 0 | 0 |
| | 7 | 736 | 105.2 | 23,146 | 0 | 0 | 0 |
| | 8 | 303 | 133.9 | 29,444 | 0 | 0 | 0 |
| | Control | | 12.2 | | 86 | 76 | 76 |
| 18 November | 1 | 303 | 47 | 19,060 | 0 | 0 | 0 |
| (1 cm) | 2 | 66 | 30 | 12,346 | 0 | 0 | 0 |
| | 3 | 65 | 25 | 10,199 | 0 | 0 | 0 |
| | 4 | 118 | 49 | 19,984 | 0 | 0 | 0 |
| | 5 | 237 | 72 | 29,242 | 0 | 0 | 0 |
| | 6 | 90 | 90 | 19,614 | 0 | 0 | 0 |
| | 7 | 129 | 85 | 34,576 | 0 | 0 | 0 |
| | 8 | 68 | 68 | 17,167 | 0 | 0 | 0 |
| | Control | | 5.2 | | 74 | 38 | 88 |
| 13 December | 1 | 88 | 36.7 | 18,735 | 0 | 0 | 0 |
| (2 cm) | 2 | 82 | 23.2 | 11,825 | 0 | 0 | 0 |
| | 3 | 122 | 53.5 | 27,260 | 0 | 0 | 0 |
| | 4 | 54 | 54.7 | 27,730 | 0 | 0 | 0 |
| | 5 | 77 | 35.6 | 18,169 | 0 | 1 | 0 |
| | 6 | 11 | 7.22 | 3684 | 84 | 73 | 85 |
| | 7 | 89 | 43.9 | 22,389 | 0 | 0 | 0 |
| | 8 | 62 | 23.7 | 12,084 | 0 | 76 | 0 |
| | Control | | 6.1 | | 86 | 59 | 86 |
| 16 December | 1 | 138 | 88.2 | 13,932 | 0 | 0 | 0 |
| (3 cm) | 2 | 146 | 80.6 | 12,729 | 0 | 0 | 0 |
| | 3 | 68 | 46.7 | 7385 | 0 | 0 | 0 |
| | 4 | 81 | 59 | 9315 | 0 | 0 | 0 |
| | 5 | 78 | 57 | 8989 | 0 | 0 | 0 |

**Table 2.** *Cont.*

| Date (Seed Location) | Plot No. | Max. Temp. (°C) | Mean Temp. (°C) | Sum of Temp. (°C) | Germination Percentages | | |
|---|---|---|---|---|---|---|---|
| | | | | | *P. mugo* | *P. contorta* | *P. sitchensis* |
| | 6 | 78 | 51.4 | 8119 | 0 | 0 | 0 |
| | 7 | 69 | 48.4 | 7646 | 0 | 71 | 0 |
| | 8 | 63 | 42.1 | 6656 | 12 | 0 | 0 |
| | Control | | 4.3 | | 82 | 42 | 48 |
| 5 April (4 cm) | 1 | 61 | 34.7 | 19,349 | 0 | 0 | 0 |
| | 2 | 84 | 45 | 24,843 | 0 | 0 | 0 |
| | 3 | 84 | 45.9 | 25,535 | 0 | 0 | 0 |
| | 4 | 43 | 20.4 | 11,066 | 70 | 0 | 63 |
| | 5 | 89 | 48.3 | 26,869 | 0 | 0 | 0 |
| | 6 | 15 | 7.9 | 4116 | 70 | 66 | 77 |
| | 7 | 28 | 14.8 | 8014 | 75 | 46 | 86 |
| | 8 | 8 | 6 | 3229 | 77 | 42 | 56 |
| | Control | | 3.8 | | 82 | 42 | 48 |
| 6 April (5 cm) | 1 | 19 | 12.6 | 6466 | 83 | 79 | 77 |
| | 2 | 61 | 28.4 | 14,550 | 0 | 10 | 0 |
| | 3 | 111 | 37.7 | 19,318 | 0 | 4 | 0 |
| | 4 | 92 | 39.0 | 20,028 | 0 | 0 | 0 |
| | 5 | 92 | 55.8 | 28,648 | 0 | 3 | 0 |
| | 6 | 76 | 35.4 | 18,137 | 0 | 7 | 0 |
| | 7 | 94 | 55.4 | 28,413 | 0 | 0 | 0 |
| | 8 | 48 | 20.8 | 10,647 | 74 | 66 | 70 |
| | Control | | 8.0 | | 70 | 34 | 58 |
| 10 April (0 cm) | 1 | 44 | 17.3 | 1509 | 82 | 52 | 66 |
| | 2 | 203 | 34.5 | 3003 | 81 | 54 | 79 |
| | 3 | 31 | 18.6 | 1619 | 72 | 42 | 62 |
| | 4 | 29 | 11.3 | 983 | 35 | 5 | 0 |
| | 5 | 17 | 8.2 | 709 | 27 | 41 | 25 |
| | 6 | 58 | 21.4 | 1857 | 10 | 49 | 25 |
| | 7 | 56 | 18.5 | 1612 | 34 | 4 | 0 |
| | 8 | 9 | 6.3 | 549 | 70 | 31 | 70 |
| | Control | | 11.3 | | 65 | 28 | 68 |

**Table 2.** *Cont.*

| Date (Seed Location) | Plot No. | Max. Temp. (°C) | Mean Temp. (°C) | Sum of Temp. (°C) | Germination Percentages | | |
|---|---|---|---|---|---|---|---|
| | | | | | *P. mugo* | *P. contorta* | *P. sitchensis* |
| 11 April | 1 | 14 | 11.2 | 559 | 80 | 48 | 86 |
| (0 cm) | 2 | 48 | 20 | 999 | 63 | 0 | 34 |
| | 3 | 11 | 9.3 | 463 | 64 | 33 | 73 |
| | 4 | 17 | 10.7 | 536 | 16 | 48 | 41 |
| | 5 | 18 | 8.9 | 447 | 71 | 38 | 42 |
| | 6 | 43 | 11.7 | 583 | 0 | 55 | 79 |
| | 7 | 81 | 20.4 | 1021 | 50 | 45 | 74 |
| | 8 | 8 | 6.6 | 331 | 77 | 37 | 61 |
| | Control | | 7.1 | | 67 | 23 | 57 |

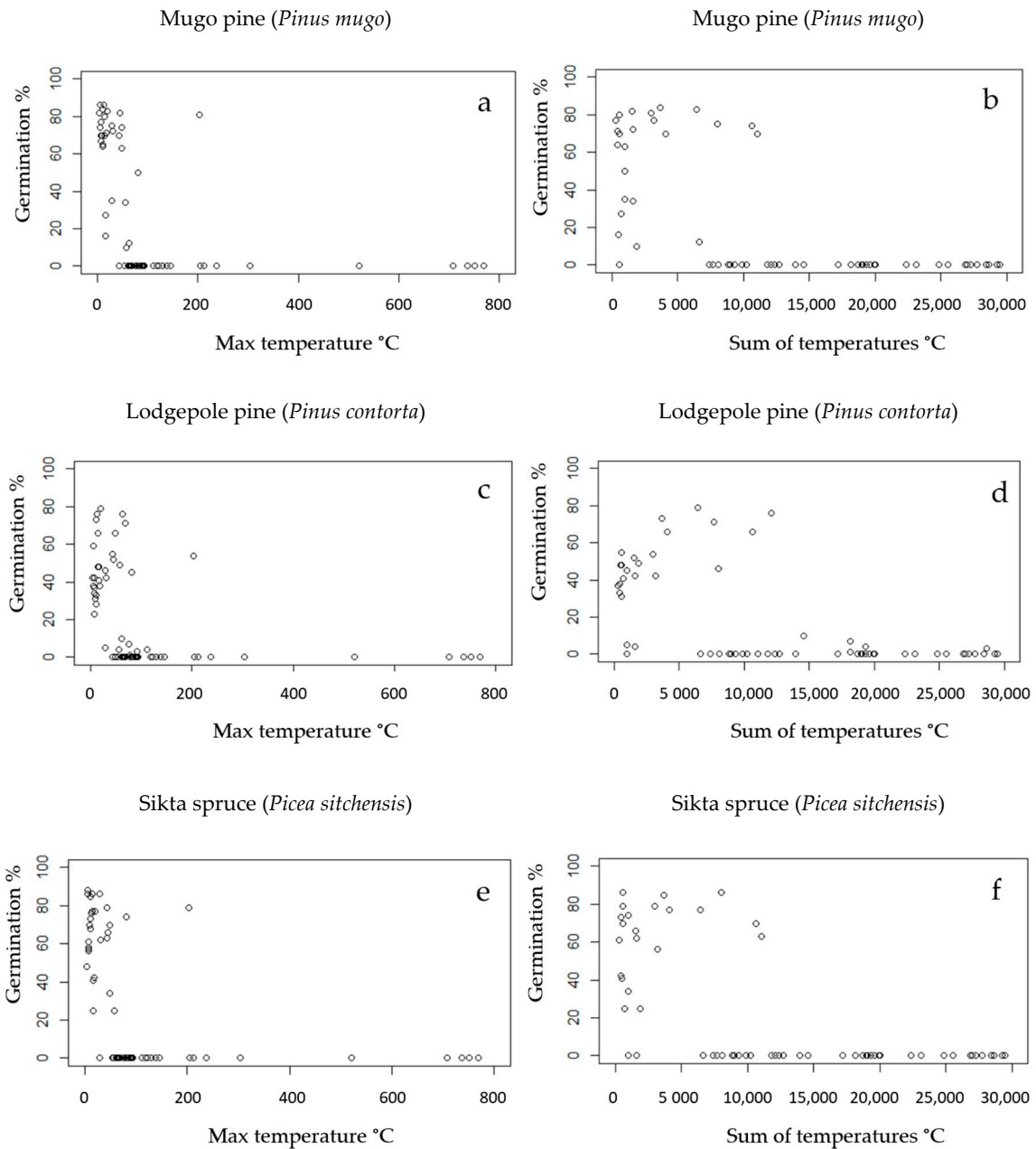

**Figure 4.** Plot of germination percentage in relation to maximum temperature measured during the burning period, and plot of germination percentage in relation to sum of temperatures based on temperature records every 30 s measure during the burning period for mugo pine (**a**,**b**), lodgepole pine (**c**,**d**), and Sitka spruce (**e**,**f**), respectively.

If we disregard one outlier at a maximum temperature at 200 °C (Figure 4), a temperature around 80 °C seems to kill the seeds of the three conifer species. The data do not allow estimating the temperature more precisely. We assume that an incorrect temperature measurement caused the outliner at 200 °C for all species.

If the sum of temperatures based on temperature records every 30 s exceeded between 12,000 and 14,000 °C, no seeds were able to germinate.

Figure 5 shows the fitted germination curves. For all three species, the quadratic terms were significant (all $p < 0.01$), supporting an inverse j-shape trend. For lodgepole pine, germination at depth 0−4 cm significantly reduced germination compared to the control ($p = 0.009$). The increase in

germination from depth 0−4 cm to 5 cm was not significant ($p = 0.08$). There was also no significant difference in germination between 5 cm and the controls ($p = 0.42$).

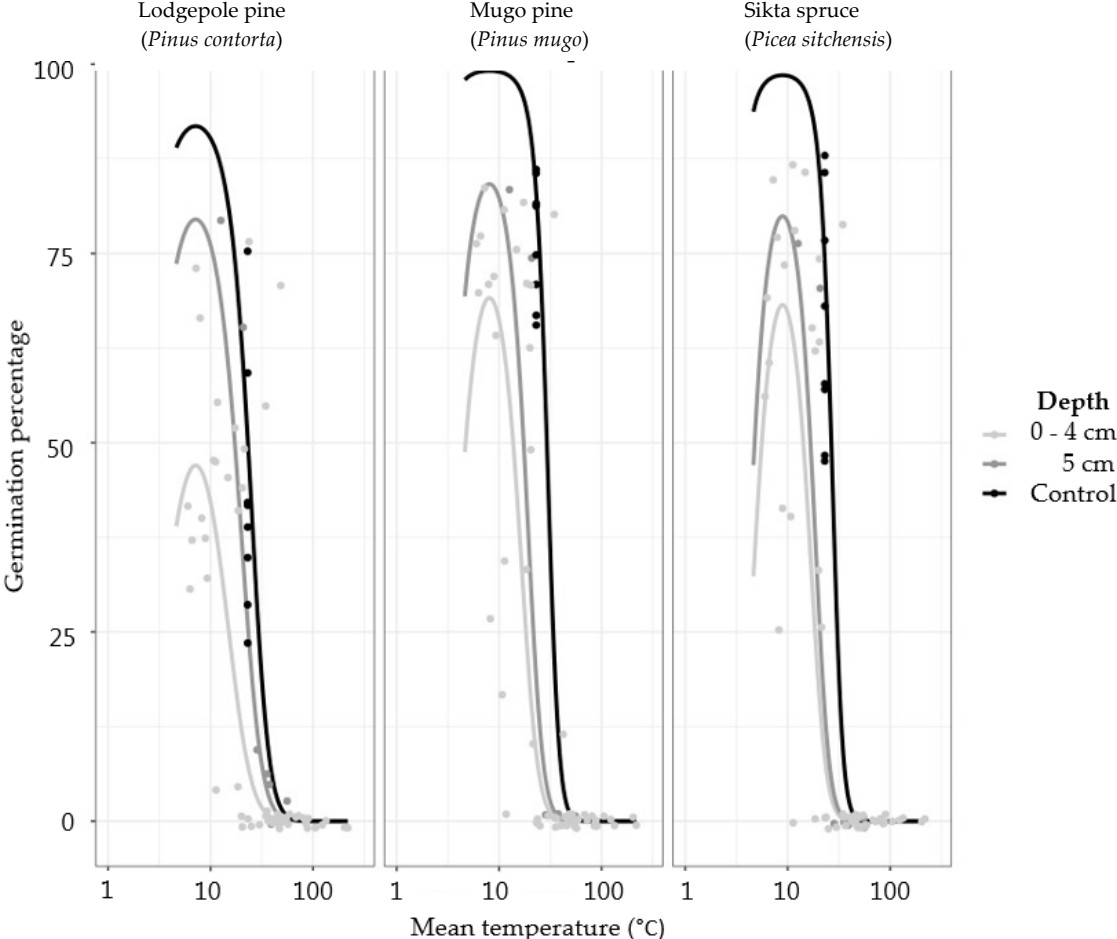

**Figure 5.** Fitted germination curves for lodgepole pine, mugo pine, and Sitka spruce, showing germination as a function of mean temperature for three different depths (0−4 cm, 5 cm, and the controls). Raw data (germination percentages) are also shown (as points).

For mugo pine, germination at depth 0−4 cm was significantly reduced compared to the control ($p < 0.0001$) whereas the increase in germination observed from depth 0−4 cm to 5 cm was not significant ($p = 0.25$). The control showed significantly higher germination compared to depth 5 cm ($p = 0.02$).

Finally, for Sitka spruce, the pattern was very similar to the one observed for mugo pine; germination at depth 0−4 cm was significantly reduced compared to the control ($p < 0.0001$) whereas the increase in germination observed from depth 0–4 cm to 5 cm was not significant ($p = 0.37$). The control group showed significantly higher germination compared to depth 5 cm ($p = 0.03$).

Seeds exposed to burning sometimes had a higher germination percentage than the controls (Table 2). Seeds were able to survive a short period of high temperatures.

## 4. Discussion

The massive clearing of lodgepole pine, Sitka spruce, and mugo pine is being carried out in an attempt to recreate and keep the original heathland in some areas in Denmark [19]. Seeds from the trees are winged and can be dispersed by wind and animals. The seeds may be buried in the soil by animals or covered with soil and sand caused by sand drift and erosion and placed in soil depths favourable for germination. Prescribed burning is a tool to control invasive conifers. We investigated the effect of prescribed burning on the seeds and found that burning of logged conifer trees often

eliminated the seeds or reduced their ability to germinate even when the seeds were placed into a depth of five centimetres in the soil. However, sometimes seeds exposed to burning seemed to germinate even better than seeds unexposed to burning. The germination might be triggered by the smoke [10–12]. Despain, Clark and Reardon [26] found that heat increased lodgepole pine seed germination to a certain extent. Seeds were tested for the ability to germinate after exposing both serotinous and non-serotinous cones for 10−60 s in a flame front designed to simulate a crown fire. Heating enhanced germination of seeds from serotinous cones but not those from non-serotinous cones. Reyes and Casal [27] studied the effect of smoke on the germination of *Pinus sylvestris* L., *Pinus nigra* Arn., *Pinus uncinata* Mill., and *P. pinaster* Ait. from Europe. They found that smoke, in the amounts tested, had no significant effects on the rate, velocity or distribution of germination, in any of the four species. Although there were no significant differences between the smoke treatments applied to the *P. pinaster* seeds, it was observed that this species had a clear tendency towards inhibition of germination as the level of smoke increased [27].

Factors such as weather conditions, humidity, and fuel load have a significant influence on the temperature of the fire and the effect of the burning, and managers should aim to achieve a high temperature for a short period to get the best result.

If the temperature exceeded a specific threshold temperature (ca. 80 °C), seeds were not able to germinate. However, the threshold temperature may vary a little between the species.

Under natural conditions, some of the seeds would probably be more or less imbibed. Tests with such seeds would give other results than tests with dried seeds. We only used dry seeds in all experiments to avoid the fact that varying imbibition levels could increase the variation in the experiment even more. Practical constraints did not allow for replicates in pace or time. For each combination of depth and species, only a single site was used (except for the control), implying that the effects of depth and site may be confounded. The consistent trend regarding depth, seen for all three species, may indicate that confounding is not severe. Still, our findings should be interpreted cautiously.

The soil layer seems to protect the seeds because the seeds buried in 5 cm depth had a higher percentage of germination. We conclude that spot firing of the conifers makes sense for clearing the area but is not sufficient as a thermal control method of the seeds. Seeds would probably be distributed all over the area and only if the temperature reaches 80 °C all seeds in a depth of 5 cm will be destroyed. The effect is depending on the sum of temperatures during the fire, which will vary a lot between and within the burning area (Figure 3). Many lodgepole pines have serotinous cones, and even if fire kills the soil seed bank of lodgepole pine it may trigger the seed release in the cones heated by the fire.

Prescribed burning also affects the fauna and surroundings. Few studies have demonstrated improvement in the conservation status of fauna resulting from prescribed burning, e.g., [28–30]. The effect on the fauna depends on the time of the year the burning takes place. For example, Radford et al. [30] found in a study from Australia that generalist rodents did not respond negatively to prescribed burning. However, two other species declined following prescribed burning in one habitat. Generalist rodents and the two declining species had a negative association with the extent of late dry season fire and a positive association with old-growth vegetation (interacting with patch size). The relative small patches of prescribed burning, which is a practice in the heathland of Denmark, would probably not have a major influence on the fauna and the natural ecosystem.

Haines et al. [31] assessed prescribed burning programs on USDA Forest Service and private and state-owned lands. Respondents to a survey reacted 14 factors for their importance as barriers to the expanding use of prescribed burning. These barriers included: (1) negative public opinion, (2) close proximity of residential development, (3) planning costs, (4) funding limitations, (5) availability of alternative silvicultural tools, (6) air quality and smoke management laws, (7) other environmental laws—excluding air quality and smoke management, (8) risk of liability for smoke intrusions and escaped fires, (9) high cost or lack of insurance availability, (10) agency or company policies that are risk-averse, (11) lack of qualified professionals and technicians, (12) excessive fuel loading, (13), a narrow

prescription window for conducting burns, and (14) uncertainty about burning as an effect fuels management practice.

## 5. Conclusions

This project showed that the burning of logged conifer trees often eliminates seeds of the three invasive conifers, even when the seeds were placed into a depth of five centimetres in the soil. However, the findings should be interpreted cautiously as each depth-species combinations were not replicated in space or time due to practical constraints. The effect of the fire on the seeds depended on the fuel load and the fire conditions. Sometimes, seeds exposed to burning seemed to germinate even better than unexposed seeds. If the sum of temperatures based on temperature records every 30 s exceeded between 12,000 and 14,000 °C, no seeds were able to germinate. In general, the conifer seeds were not able to germinate if they were exposed to a temperature of around 80 °C. Consequently, managers should strive for obtaining a high temperature exceeding this threshold and keep it for a while to achieve the desired reduction of the soil seed bank of the three invasive conifer species.

**Author Contributions:** K.R. contributed with the conceptualization, methodology and investigation supervised by C.A., K.R. and C.R. made formal analysis. C.R. did the statistical analyses and described the statistical method. C.A. wrote the original draft. C.A., K.R. and C.R. reviewed and edited the manuscript. All authors have read and agreed to the published version of the manuscript.

**Funding:** We thank Frederik Kølpin Ravn's and Ernst Gram Trust Fond and the Obelske Family Trust Fond for supporting to work.

**Acknowledgments:** We thank Forest Superintendent Frede Jensen and his staff at Danish Forest and Nature Agency, Forest District of Northern Jutland, Skagen, for their support throughout the planning and execution of this study. We also thank Anders Drustrup. Danish Institute of Fire and Security for providing thermocouples and for valuable suggestions and recommendations regarding the experimental design.

**Conflicts of Interest:** The authors declare no conflict of interest. The funders had no role in the design of the study, in the collection, analyses or interpretation of data; in the writing of the manuscript or in the decision to publish the results.

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
