# Peer review of "Prescribed Burning as A Management Tool to Destroy Dry Seeds of Invasive Conifers in Heathland in Denmark"

_land, doi:10.3390/land9110432_

Round 1
Reviewer 1 Report
The authors are describing the effects of prescribed fire on the seed of three invasive conifers - Pinus mugo, lodgepole pine, and Sitka spruce. They attempted to replicate the field conditions under which the seed might be exposed to fire, which resulted in a variety of temperatures to which the seed was exposed.
Line 38 should be "threat" not "treat"
The literature review/introduction does not address any literature/studies related to the specific species under study. There should be data available (perhaps in the Woody Plant Seed Manual?) about longevity of seed in the soil seed bank and the deepest depth at which the seed of that species will germinate. There may also be information available about some of these species in the Fire Effects Information System online.
In the methods, any pretreatment of the seed prior to the testing is not described (if any). During the time of year when these burning tests were conducted, I would expect that the seed would naturally be imbibed. Tests with imbibed seed may (and probably will definitely have) a different result than tests with dried seed. I suspect that the seed was not imbibed in advance from the description of the germination tests. If the seed moisture content was lower than would naturally occur during this time of year, that should be dealt with in the discussion section of the paper.
I am somewhat confused about the "Statistical Analysis" section in the methods. The first line indicates that the response of each seed is binary. But the experimental unit is not an individual seed - the experimental unit is the group of seeds within the wire mesh bag.
I think the presentation of results could be simplified. The temperature sum was calculated, but the only place it appears to be used is in the lengthy table that displays the germination percentages for each species for each plot for each day's test. It could be interesting and informative to plot germination vs. the temperature sum for each tree species as well as to plot germination vs. maximum temperature for each species. The latter would help support the temperature thresholds presented in the conclusions.
Some of the statements in the conclusions are confusing to me.
Line 253-254: Seeds were able to survive a short period of high temperatures.
Line 255-256: If the temperature exceeded the threshold, the seeds were not able to germinate afterwards.
These 2 statements appear to conflict - how long is a "short period"? And how does "short period of high temperatures" relate to the threshold. I suggest deleting lines 253-254. There doesn't appear to be any data presented in the results to support Line 253-254.
The discussion also talks about the mean temperature, but if you are looking for a threshold, I am not sure why the mean temperature would be important. Usually the key thing in these types of studies is the temperature threshold and the associated time spent at that threshold to generate the germination results.
Author Response
Thanks a lot for the many usful comments and suggestions. We have answers below in bold.
The authors are describing the effects of prescribed fire on the seed of three invasive conifers - Pinus mugo, lodgepole pine, and Sitka spruce. They attempted to replicate the field conditions under which the seed might be exposed to fire, which resulted in a variety of temperatures to which the seed was exposed.
Line 38 should be "threat" not "treat"
Corrected
The literature review/introduction does not address any literature/studies related to the specific species under study. There should be data available (perhaps in the Woody Plant Seed Manual?) about longevity of seed in the soil seed bank and the deepest depth at which the seed of that species will germinate.
There may also be information available about some of these species in the Fire Effects Information System online.
We have only succeed in finding very little information about seed viability and longevity of seeds for the three species. We have added the information in the introduction.
In the methods, any pretreatment of the seed prior to the testing is not described (if any).
Yes, it was mentioned from line 158
During the time of year when these burning tests were conducted, I would expect that the seed would naturally be imbibed. Tests with imbibed seed may (and probably will definitely have) a different result than tests with dried seed. I suspect that the seed was not imbibed in advance from the description of the germination tests. If the seed moisture content was lower than would naturally occur during this time of year, that should be dealt with in the discussion section of the paper.
We have changed the headline and added to the discussion: Under natural conditions, some of the seeds would probably be more or less imbibed. Tests with such seeds would give other results than tests with dried seeds. We only used dry seeds in all experiments to avoid varying imbibition level could increase the variation in the experiment even more.
I am somewhat confused about the "Statistical Analysis" section in the methods. The first line indicates that the response of each seed is binary. But the experimental unit is not an individual seed - the experimental unit is the group of seeds within the wire mesh bag.
Germination data are binary data (germination; no germination) and should always be treated as such. Consequently, we also need to consider tht when we work with germinating populations of seeds. The population does not follow a normal distribution and we have taken that into account in the model we used.
I think the presentation of results could be simplified. The temperature sum was calculated, but the only place it appears to be used is in the lengthy table that displays the germination percentages for each species for each plot for each day's test. It could be interesting and informative to plot germination vs. the temperature sum for each tree species as well as to plot germination vs. maximum temperature for each species. The latter would help support the temperature thresholds presented in the conclusions.
Thanks a lot for this good suggestion. We agree it gives a much better overview of the results. We have now made the figures as recommended to give a better overview of the results. We have also moderate the statements about temperature thresholds all through the paper.
However, we would like to keep the Table 2 because it shows the relation between buried depths of the seeds and germination. The other reviewers were also pleased with Table 2.
Some of the statements in the conclusions are confusing to me.
Line 253-254: Seeds were able to survive a short period of high temperatures.
We have deleted the sentence
Line 255-256: If the temperature exceeded the threshold, the seeds were not able to germinate afterwards. These 2 statements appear to conflict - how long is a "short period"? And how does "short period of high temperatures" relate to the threshold.
I suggest deleting lines 253-254. There doesn't appear to be any data presented in the results to support Line 253-254.
We have deleted the sentence.
The discussion also talks about the mean temperature, but if you are looking for a threshold, I am not sure why the mean temperature would be important. Usually the key thing in these types of studies is the temperature threshold and the associated time spent at that threshold to generate the germination results.
We agree in this, and we have stressed that in the rewritten conclusion.
Reviewer 2 Report
The article sufficiently describes the effect of fire or high temperatures on the seeds of invasive conifers. However, this does not sufficiently address the aspect of management and fire as tools. Therefore, it is necessary to either change the title or modify the content of the article to include other aspects of the use of fire as a management tool. For example, what effect does fire have on the rest of the organisms. Or under what conditions fire can be used as a management tool. These aspects would be well commented on in the discussion.
Author Response
Thank you for your suggestion. Please find below our answers in bold text
The article sufficiently describes the effect of fire or high temperatures on the seeds of invasive conifers. However, this does not sufficiently address the aspect of management and fire as tools. Therefore, it is necessary to either change the title or modify the content of the article to include other aspects of the use of fire as a management tool. For example, what effect does fire have on the rest of the organisms. Or under what conditions fire can be used as a management tool. These aspects would be well commented on in the discussion.
We have changes the headline. From the headline, it should be clear that we are only focusing on prescribed burning as a tool to destroy seeds. The paper does not concerns other organisms. We have changes the headline to “Prescribed burning as a management tool to destroy seeds of…...”
However, we agree that it is relevant to mention other effects on the environments and barriers for using prescribed burning in the discussion. We have now added two new paragraphs:
Prescribed burning also effects the fauna and the surroundings. Few studies have demonstrated improvements in the conservation status of fauna resulting from prescribed burning (e.g., Clarke 2008; Driscoll et al. 2010, Radford et al. 2020). The effect on the fauna would depend on the time of the year the burning takes place. For example, Radford et al. [] found in a study from Australia that generalist rodents did not respond negatively to prescribed burning. However, two other species declined following prescribed burning in one habitat. Generalist rodents and the two declining species had a negative association with extent of late dry season fire and a positive association with old growth vegetation (interacting with patch size). The relatively small patches of prescribed burning which is practice in the heathland in Denmark would probably not have a major influence on the ecosystems.
Haines at al. [] assessed prescribed burning programs on USDA Forest Service and private and state-owned lands. Respondents to a survey rated 14 factors for their importance as barriers to the expanding the use of prescribed burning. These barriers included: (1) negative public opinion, (2) close proximity of residential development, (3) planning costs, (4) funding limitations, (5) availability of alternative silvicultural tools, (6) air quality and smoke management laws, (7) other environmental laws—excluding air quality and smoke management, (8) risk of liability for smoke intrusions and escaped fires, (9) high cost or lack of insurance availability, (10) agency or company policies that are risk-averse, (11) lack of qualified professionals and technicians, (12) excessive fuel loading, (13) a narrow prescription window for conducting burns, and (14) uncertainty about burning as an effect fuels management practice .
Reviewer 3 Report
It has to be emphasised that this paper is interesting and deals with relevant issues. I do not have many suggestions but I have a few essential ones.
- Sustainable development definition has to be added, especially in the context of land restoration.
- Conclusions party practically does not exists. Plaese expend conclusions with valuable practical findings and foresight.
- References are very poor at this moment. Please use much more proper sources from the very rich field of reseach.
Author Response
Thanks for the comments. Please find our answers below in bold text.
It has to be emphasised that this paper is interesting and deals with relevant issues. I do not have many suggestions but I have a few essential ones.
- Sustainable development definition has to be added, especially in the context of land restoration.
We have not mentioned “Sustainable development” in the manuscript before. If we understand the reviewer correctly, we should do so and define it.
We have added:
“Prescribed burning is often considered a sustainable management tool in land restoration because fires occurs naturally in dry landscapes and contribute to a sustainable development of natural ecosystems; meaning a development that meets the needs of the present without compromising the ability of future generations of native species to meet their needs”.
- Conclusions party practically does not exists. Plaese expend conclusions with valuable practical findings and foresight.
We have changed the conclusion:
This project showed that burning of logged conifer trees often eliminate seeds of the three invasive conifers even when the seeds were placed into a depth of five centimetres in the soil. However, findings should be interpreted cautiously as each depth-species combinations were not replicated in space or time due to practical constraints. The effect of the fire on the seeds depended on the fuel load and the fire conditions. Sometimes seeds exposed to burning seemed to germinate even better than unexposed seeds. If the sum of temperatures based on temperature records every 30 seconds exceeded between 12000 and 14000 °C no seeds were able to germinate. In general, the conifer seeds were not able to germinate if they were exposed to a temperature of around 80 °C. Consequently, managers should strive for obtaining a high temperature exceeding this threshold and keep it for a while to achieve the desired reduction of the soil seed bank of the three invasive conifer species
- References are very poor at this moment. Please use much more proper sources from the very rich field of reseach.
We have extended the introduction and discussion and added more references